# Learning Finite State Representations of Recurrent Policy Networks

**Anurag Koul & Alan Fern**
School of EECS
Oregon State University
Corvallis, Oregon, USA
{koula,alan.fern}@oregonstate.edu

**Sam Greydanus** [*]
Google Brain
Mountain View, California, USA
sgrey@google.com

## Abstract

Recurrent neural networks (RNNs) are an effective representation of control policies for a wide range of reinforcement and imitation learning problems. RNN policies, however, are particularly difficult to explain, understand, and analyze due to their use of continuous-valued memory vectors and observation features. In this paper, we introduce a new technique, Quantized Bottleneck Insertion, to learn finite representations of these vectors and features. The result is a quantized representation of the RNN that can be analyzed to improve our understanding of memory use and general behavior. We present results of this approach on synthetic environments and six Atari games. The resulting finite representations are surprisingly small in some cases, using as few as 3 discrete memory states and 10 observations for a perfect Pong policy. We also show that these finite policy representations lead to improved interpretability.

## 1 Introduction

Deep reinforcement learning (RL) and imitation learning (IL) have demonstrated impressive performance across a wide range of applications. Unfortunately, the learned policies are difficult to understand and explain, which limits the degree that they can be trusted and used in high-stakes applications. Such explanations are particularly problematic for policies represented as recurrent neural networks (RNNs) (Mnih et al., 2016; Mikolov et al., 2010), which are increasingly used to achieve state-of-the-art performance (Mnih et al., 2015; Silver et al., 2017). This is because RNN policies use internal memory to encode features of the observation history, which are critical to their decision making, but extremely difficult to interpret. In this paper, we take a step towards comprehending and explaining RNN policies by learning more compact memory representations.

Explaining RNN memory is challenging due to the typical use of high-dimensional continuous memory vectors that are updated through complex gating networks (e.g. LSTMs, GRUs (Hochreiter & Schmidhuber, 1997; Chung et al., 2014; Cho et al., 2014)). We hypothesize that, in many cases, the continuous memory is capturing and updating one or more discrete concepts. If exposed, such concepts could significantly aid explainability. This motivates attempting to quantize the memory and observation representation used by an RNN to more directly capture those concepts. In this case, understanding the memory use can be approached by manipulating and analyzing the quantized system. Of course, not all RNN policies will have compact quantized representations, but many powerful forms of memory usage can be captured in this way.

Our main contribution is to introduce an approach for transforming an RNN policy with continuous memory and continuous observations to a finite-state representation known as a Moore Machine. To accomplish this we introduce the idea of *Quantized Bottleneck Network (QBN) insertion*. QBNs are simply auto-encoders, where the latent representation is quantized. Given a trained RNN, we train QBNs to encode the memory states and observation vectors that are encountered during the RNN operation. We then insert the QBNs into the trained RNN policy in place of the "wires" that propagated the memory and observation vectors. The combination of the RNN and QBN results in a

---

[*]Work done at Oregon State University

policy represented as a Moore Machine Network (MMN) with quantized memory and observations that is nearly equivalent to the original RNN. The MMN can be used directly or fine-tuned to improve on inaccuracies introduced by QBN insertion.

While training quantized networks is often considered to be quite challenging, we show that a simple approach works well in the case of QBNs. In particular, we demonstrate that "straight through" gradient estimators as in (Bengio et al., 2013; Courbariaux et al., 2016) are quite effective.

We present experiments in synthetic domains designed to exercise different types of memory use as well as benchmark grammar learning problems. Our approach is able to accurately extract the ground-truth MMNs, providing insight into the RNN memory use. We also did experiments on 6 Atari games using RNNs that achieve state-of-the-art performance. We show that in most cases it is possible to extract near-equivalent MMNs and that the MMNs can be surprisingly small. Further, the extracted MMNs give insights into the memory usage that are not obvious based on just observing the RNN policy in action. For example, we identify games where the RNNs do not use memory in a meaningful way, indicating the RNN is implementing purely reactive control. In contrast, in other games, the RNN does not use observations in a meaningful way, which indicates that the RNN is implementing an open-loop controller.

## 2 RELATED WORK

There have been efforts made in the past to understand the internals of Recurrent Networks (Karpathy et al., 2015; Arras et al., 2017; Strobelt et al., 2016; Murdoch & Szlam, 2017; Jacobsson, 2005; Omlin & Giles, 1996). However, to the best of our knowledge there is no prior work on learning finite-memory representations of continuous RNN policies. Our work, however, is related to a large body of work on learning finite-state representations of recurrent neural networks. Below we summarize the branches of that work and the relationship to our own.

There has been a significant history of work on extracting Finite State Machines (FSMs) from recurrent networks trained to recognize languages (Zeng et al., 1993; Tiňo et al., 1998; Cechin et al., 2003). Typical approaches include discretizing the continuous memory space via gridding or clustering followed by minimization. A more recent approach is to use classic query-based learning algorithms to extract FSMs by asking membership and equivalence queries (Weiss et al., 2017). However, none of these approaches directly apply to learning policies, which require extending to Moore Machines. In addition, all of these approaches produce an FSM approximation that is separated from the RNN and thus serve as only a proxy of the RNN behavior. Rather, our approach directly inserts discrete elements into the RNN that preserves its behavior, but allows for a finite state characterization. This insertion approach has the advantage of allowing fine-tuning and visualization using standard learning frameworks.

The work most similar to ours also focused on learning FSMs (Zeng et al., 1993). However, the approach is based on directly learning recurrent networks with finite memory, which are qualitatively similar to the memory representation of our MMNs. That work, however, focused on learning from scratch rather than aiming to describe the behavior of a continuous RNN. Our work extends that approach to learn MMNs and more importantly introduces the method of QBN insertion as a way of learning via guidance from a continuous RNN.This transforms any pre-trained recurrent policy into a finite representation.

We note that there has been prior work on learning fully binary networks, where the activation functions and/or weights are binary (e.g. (Bengio et al., 2013; Courbariaux et al., 2016; Hinton, 2012)). The goal of that line of work is typically to learn more time and space efficient networks. Rather, we focus on learning only discrete representations of memory and observations, while allowing the rest of the network to use arbitrary activations and weights. This is due to our alternative goal of supporting interpretability rather than efficiency.

## 3 RECURRENT POLICY NETWORKS: CONTINUOUS AND QUANTIZED

Recurrent neural networks (RNNs) are commonly used in reinforcement learning to represent policies that require or can benefit from internal memory. At each time step, an RNN is given an observation $o_t$ (e.g. image) and must output an action $a_t$ to be taken in the environment. During

execution an RNN maintains a continuous-valued hidden state $h_t$, which is updated on each transition and influences the action choice. In particular, given the current observation $o_t$ and current state $h_t$, an RNN performs the following operations: 1) Extract a set of observation features $f_t$ from $o_t$, for example, using a CNN when observations are images, 2) Outputting an action $a_t = \pi(h_t)$ according to policy $\pi$, which is often a linear softmax function of $h_t$, 3) transition to a new state $h_{t+1} = \delta(f_t, h_t)$ where $\delta$ is the transition function, which is often implemented via different types of gating networks such as LSTMs or GRUs.

The continuous and high dimensional nature of $h_t$ and $f_t$ can make interpreting the role of memory difficult. This motivates our goal of extracting compact quantized representations of $h_t$ and $f_t$. Such representations have the potential to allow investigating a finite system that captures the key features of the memory and observations. For this purpose we introduce Moore Machines and their deep network counterparts.

**Moore Machines.** A classical *Moore Machine (MM)* is a standard finite state machine where all states are labeled by output values, which in our case will correspond to actions. In particular, a Moore Machine is described by a finite set of (hidden) states $\hat{H}$, an initial hidden state $\hat{h}_0$, a finite set of observations $\hat{O}$, a finite set of actions $A$, a transition function $\hat{\delta}$, and a policy $\hat{\pi}$ that maps hidden states to actions. The transition function $\hat{\delta} : \hat{H} \times \hat{O} \rightarrow \hat{H}$ returns the next hidden state $\hat{h}_{t+1} = \hat{\delta}(\hat{h}_t, \hat{o}_t)$ given the current state $\hat{h}_t$ and observation $\hat{o}_t$. By convention we will use $h_t$ and $\hat{h}_t$ to denote continuous and discrete states respectively and similarly for other quantities and functions.

**Moore Machine Networks.** A *Moore Machine Network (MMN)* is a Moore Machine where the transition function $\hat{\delta}$ and policy $\hat{\pi}$ are represented via deep networks. In addition, since the raw observations given to an MMN are often continuous, or from an effectively unbounded set (e.g. images), an MMN will also provide a mapping $\hat{g}$ from the continuous observations to a finite discrete observation space $\hat{O}$. Here $\hat{g}$ will also be represented via a deep network. In this work, we consider quantized state and observation representations where each $\hat{h} \in \hat{H}$ is a discrete vector and each discrete observation in $\hat{O}$ is a discrete vector that describes the raw observation. We will denote the quantization level as $k$ and the dimensions of $\hat{h}$ and $\hat{f}$ by $B_h$ and $B_f$ respectively.

Based on the above discussion, an MMN can be viewed as a traditional RNN, where: 1) The memory is restricted to be composed of $k$-level activation units, and 2) The environmental observations are intermediately transformed to a $k$-level representation $\hat{f}$ before being fed to the recurrent module. Given an approach for incorporating quantized units into the backpropagation process, it is straightforward, in concept, to learn MMNs from scratch via standard RNN learning algorithms. However, we have found that learning MMNs from scratch can be quite difficult for non-trivial problems, even when an RNN can be learned with relative ease. For example, we have not been able to train high-performing MMNs from scratch for Atari games. Below we introduce a new approach for learning MMNs that is able to leverage the ability to learn RNNs.

## 4 LEARNING MOORE MACHINE NETWORKS FROM RNNS

Given a trained RNN, our key idea is to first learn *quantized bottleneck networks (QBNs)* for embedding the continuous observation features and hidden state into a $k$-level quantized representation. We will then insert the QBNs into the original recurrent net in such a way that its behavior is minimally changed with the option of fine-tuning after insertion. The resulting network can be viewed as consuming quantized features and maintaining quantized state, which is effectively an MMN. Below we describe the steps in further detail, which are illustrated in Figure 1.

### 4.1 QUANTIZED BOTTLENECK NETWORKS

A QBN is simply an autoencoder where the latent representation between the encoder and decoder (i.e. the bottleneck) is constrained to be composed of $k$-level activation units. While, traditional autoencoders are generally used for the purpose of dimensionality reduction in continuous space (Hinton & Salakhutdinov, 2006), QBNs are motivated by the goal of discretizing a continuous space. Conceptually, this can be done by quantizing the activations of units in the encoding layer.

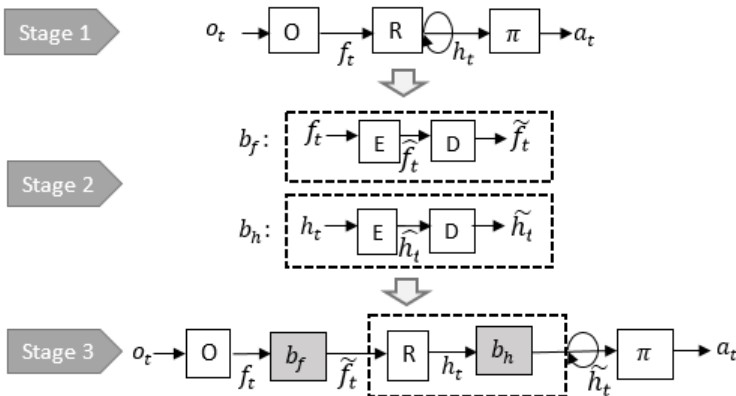

Figure 1: Learning Moore Machine Networks. (1.) Learn an RNN policy. (2.) Learn QBN's to quantize memory and observations. (3.) Insertion and Fine Tuning. The modules labelled O, R are observation feature-extraction and recurrent modules, respectively

We represent a QBN via a continuous multilayer encoder $E$, which maps inputs $x$ to a latent encoding $E(x)$, and a corresponding multilayer decoder $D$. To quantize the encoding, the QBN output is given by
$$b(x) = D(\text{quantize}(E(x)))$$
In our case, we use 3-level quantization in the form of $+1, 0$ and $-1$ using the quantize function, which assumes the outputs of $E(x)$ are in the range $[-1, 1]$.[1] One choice for the output nodes of $E(x)$ would be the $\tanh$ activation. However, since the gradient of $\tanh$ is close to 1 near 0, it can be difficult to produce quantization level 0 during learning. Thus, as suggested in Pitis (2017), to support 3-valued quantization we use the following activation function, which is flatter in the region around zero input.
$$\phi(x) = 1.5 \tanh(x) + 0.5 \tanh(-3x)$$
Of course introducing the *quantize* function in the QBN results in $b(x)$ being non-differentiable, making it apparently incompatible with backpropagation, since the gradients between the decoder and encoder will almost always be zero. While there are a variety of ways to deal with this issue, we have found that the straight-through estimator, as suggested and used in prior work (Hinton, 2012; Bengio et al., 2013; Courbariaux et al., 2016) is quite effective. In particular, the standard straight-through estimator of the gradient simply treats the quantize function as the identity function during back-propagation. Overall, the inclusion of the quantize function in the QBN effectively allows us to view the last layer of $E$ as producing a $k$-level encoding. We train a QBN as an autoencoder using the standard $L_2$ reconstruction error $\|x - b(x)\|^2$ for a given input $x$.

## 4.2 BOTTLENECK INSERTION

Given a recurrent policy we can run the policy in the target environment in order to produce an arbitrarily large set of training sequences of triples $(o_t, f_t, h_t)$, giving the observation, corresponding observation feature, and hidden state at time $t$. Let $F$ and $H$ be the sets of all observed features and states respectively. The first step of our approach is to train two QBNs, $b_f$ and $b_h$, on $F$ and $H$ respectively. If the QBNs are able to achieve low reconstruction error then we can view latent "bottlenecks" of the QBNs as a high-quality $k$-level encodings of the original hidden states and features.

We now view $b_f$ and $b_h$ as "wires" that propagate the input to the output, with some noise due to imperfect reconstruction. We insert these wires into the original RNN in the natural way (stage-3 in Figure 1). The $b_f$ QBN is inserted between the RNN units that compute the features $f$ and the nodes those units are connected to. The $b_h$ QBN is inserted between the output of the recurrent network block and the input to the recurrent block. If $b_f$ and $b_h$ always produced perfect reconstructions, then

---

[1]This three valued approach was chosen for its potential interpretability of a quantized output having negative, positive, or zero influence. Other quantizations are worth future exploration.

the result of inserting them as described would not change the behavior of the RNN. Yet, the RNN can now be viewed as an MMN since the bottlenecks of $b_f$ and $b_h$ provide a quantized representation of the features $f_t$ and states $h_t$.

**Fine Tuning.** In practice, the QBNs will not achieve perfect reconstruction and thus, the resulting MMN may not behave identically to the original RNN. Empirically, we have found that the performance of the resulting MMN is often very close to the RNN directly after insertion. However, when there is non-trivial performance degradation, we can fine-tune the MMN by training on the original rollout data of the RNN. Importantly, since our primary goal is to learn a representation of the original RNN, during fine-tuning our objective is to have the MMN match the softmax distribution over actions produced by the RNN. We found that training in this way was significantly more stable than training the MMN to simply output the same action as the RNN.

### 4.3 MOORE MACHINE EXTRACTION AND MINIMIZATION

After obtaining the MMN, one could use visualization and other analysis tools to investigate the memory and it's feature bits in order to gain a semantic understanding of their roles. Solving the full interpretation problem in a primitive way is beyond the scope of this work.

**Extraction.** Another way to gain insight is to use the MMN to produce an equivalent Moore Machine over atomic state and observation spaces, where each state and observation is a discrete symbol. This machine can be analyzed to understand the role of different machine states and how they are related. In order to create the Moore Machine we run the learned MMN to produce a dataset of $< \hat{h}_{t-1}, \hat{f}_t, \hat{h}_t, a_t >$, giving the consecutive pairs of quantized states, the quantized features that led to the transition, and the action selected after the transition. The state-space of the Moore Machine will correspond to the $p$ distinct quantized states in the data and the observation-space of the machine will be the $q$ unique quantized feature vectors in the data. The transition function of the machine $\hat{\delta}$ is constructed from the data by producing a $p \times q$ transaction table that captures the transitions seen in the data.

**Minimization.** In general, the number of states $p$ in the resulting Moore Machine will be larger than necessary in the sense that there is a much smaller, but equivalent, minimal machine. Thus, we apply standard Moore Machine minimization techniques to arrive at the minimal[2] equivalent Moore Machine (Paull & Unger, 1959). This often dramatically reduces the number of distinct states and observations.

## 5 SYNTHETIC EXPERIMENTS

Our experiments address the following questions: 1) Is it possible to extract MMNs from RNNs without significant loss in Performance? 2) What is the general magnitude of the number of states and observations in the minimal machines, especially for complex domains such as Atari? 3) Do the learned MMNs help with interpretability of the recurrent policies? In this section, we begin addressing these questions by considering two domains where ground truth Moore Machines are known. The first is a parameterized synthetic environment, Mode Counter, which can capture multiple types of memory use. Second, we consider benchmark grammar learning problems.

### 5.1 MODE COUNTER

The class of *Mode Counter Environments (MCEs)* allows us to vary the amount of memory required by a policy (including no memory) and the required type of memory usage. In particular, MCEs can require varying amounts of memory for remembering past events and implementing internal counters.

An MCE is a restricted type of Partially Observable Markov Decision Process, which transitions between one of $M$ modes over time according to a transition distribution, which can depend on

---

[2]Typically minimization algorithms focus on minimizing only the number of states and do not change the number of observations. We also aim to minimize the observations. In particular, if there are observations that have identical transition dynamics, then we merge them into one observation. This preserves equivalence to the original machine, but can dramatically decrease the number of required observation symbols.

the current mode and amount of time spent in the current mode. There are $M$ actions, one for each mode, and the agent receives a reward of $+1$ at the end of the episode if it takes the correct action associated with the active mode at each time step. The agent does not observe the mode directly, but rather must infer the mode via a combination of observations and memory use. Different parameterizations place different requirements on how (and if) memory needs to be used to infer the mode and achieve optimal performance. Below we give an intuitive description of the MCEs[3] in our experiments.

We conduct experiments in three MCE instances, which use memory and observations in fundamentally different ways. This tests our ability to use our approach for determining the type of memory use. **1) Amnesia.** This MCE is designed so that the optimal policy does not need memory to track past information and can select optimal actions based on just the current observation. **2) Blind.** Here we consider the opposite extreme, where the MCE observations provide no information about optimal actions. Rather, memory must be used to implement counters that keep track of a deterministic mode sequence for determining the optimal actions. **3) Tracker.** This MCE is designed so that the optimal policy must both use observations and memory in order to select optimal actions. Intuitively the memory must implement counters that keep track of key time steps where the observations provide information about the mode. In all above instances, we used $M = 4$ modes.

**RNN Training.** For each MCE instance we use the following recurrent architecture: the input feeds into 1 feed-forward layer with 4 Relu6 nodes (Krizhevsky & Hinton, 2010) ($f_t$), followed by a 1-layer GRU with 8 hidden units ($h_t$), followed by a fully connected softmax layer giving a distribution over the $M$ actions (one per mode). Since we know the optimal policy in the MCEs we use imitation learning for training. For all of the MCEs in our experiments, the trained RNNs achieve 100% accuracy on the imitation dataset and appeared to produce optimal policies.

**MMN Training.** The observation QBN $b_f$ and hidden-state QBN $b_h$ have the same architecture, except that the number of quantized bottleneck units $B_f$ and $B_h$ are varied in our experiments. The encoders consist of 1 feed-forward layer of $\tanh$ nodes, where the number of nodes is 4 times the size of the bottleneck. This layer feeds into 1 feedforward layer of quantized bottleneck nodes (see Section 4). The decoder for both $b_f$ and $b_h$ has a symmetric architecture to the encoder. Training of $b_f$ and $b_h$ in the MCE environments was extremely fast compared to the RNN training, since QBNs do not need to learn temporal dependencies. We trained QBNs with bottleneck sizes of $B_f \in \{4, 8\}$ and $B_h \in \{4, 8\}$. For each combination of $B_f$ and $B_h$ we embedded the QBNs into the RNN to give a discrete MMN and then measured performance of the MMN before and after fine tuning. Table 1 gives the average test score over 50 test episodes. Score of 1 indicates the agent performed optimally for all episodes. In most of the cases no fine tuning was required (marked as '-') since the agent achieved perfect performance immediately after bottleneck insertion due to low reconstruction error. In all other cases, except for Tracker ($B_h = 4$, $B_f = 4$) fine-tuning resulted in perfect MMN performance. The exception yielded 98% accuracy. Interestingly in that case, we see that if we only insert one of the bottlenecks at a time, we yield perfect performance, which indicates that the combined error accumulation of the two bottlenecks is responsible for the reduced performance.

**Moore Machine Extraction.** Table 1 also gives the number of states and observations of the MMs extracted from the MMNs both before and after minimization. Recall that the number of states and obsevations before minimization is the number of distinct combinations of values observed for the bottleneck nodes during long executions of the MMN. We see that there are typically significantly more states and observations before minimization than after. This indicates that the MMN learning does not necessarily learn minimal discrete state and observation representations, though the representations accurately describe the RNN. After minimization (Section 4.3), however, in all but one case we get exact minimal machines for each MCE domain. The ground truth minimal machines that are found are shown in the Appendix (Figure 3). This shows that the MMNs learned via QBN insertions were equivalent to the true minimal machines and hence indeed optimal in most cases. The exception matches the case where the MMN did not achieve perfect accuracy.

Examining these machines allows one to understand the memory use. For example, the machine for Blind has just a single observation symbol and hence its transitions cannot depend on the input observations. In contrast, the machine for Amnesia shows that each distinct observation symbol

---

[3]Appendix has full details of how MCEs are parameterized and the specific instances used in our experiments.

leads to the same state (and hence action choice) regardless of the source state. Thus, the policies action is completely determined by the current observation.

Table 1: Moore Machine extraction for MCE

| Game | $B_h, B_f$ | Fine-Tuning Score | | Before Minimization | | | After Minimization | | |
|------|------------|-------------------|-------------|----------------------|-------------|------------|--------------------|-------------|------------|
| | | Before (%) | After (%) | $\left\|\hat{H}\right\|$ | $\left\|\hat{O}\right\|$ | Acc. (%) | $\left\|\hat{H}\right\|$ | $\left\|\hat{O}\right\|$ | Acc. (%) |
| Amnesia | 4,4 | 0.98 | 1 | 7 | 5 | 1 | **4** | **4** | 1 |
| | 4,8 | 0.99 | 1 | 7 | 7 | 1 | **4** | **4** | 1 |
| | 8,4 | 1 | - | 6 | 5 | 1 | **4** | **4** | 1 |
| | 8,8 | 0.99 | 1 | 7 | 7 | 1 | **4** | **4** | 1 |
| Blind | 4,4 | 1 | - | 12 | 6 | 1 | **10** | **1** | 1 |
| | 4,8 | 1 | - | 12 | 8 | 1 | **10** | **1** | 1 |
| | 8,4 | 1 | - | 5 | 6 | 1 | **10** | **1** | 1 |
| | 8,8 | 0.78 | 1 | 13 | 8 | 1 | **10** | **1** | 1 |
| Tracker | 4,4 | 0.98 | 0.98 | 58 | 5 | 0.98 | **50** | **4** | 0.98 |
| | 4,8 | 0.99 | 1 | 23 | 5 | 1 | **10** | **4** | 1 |
| | 8,4 | 0.98 | 1 | 91 | 5 | 1 | **10** | **4** | 1 |
| | 8,8 | 0.99 | 1 | 85 | 5 | 1 | **10** | **4** | 1 |

## 5.2 TOMITA GRAMMARS

The Tomita Grammars are popular benchmarks for learning finite state machines (FSMs), including work on extracting FSMs from RNNs (e.g. (Watrous & Kuhn, 1992; Weiss et al., 2017)). Here we evaluate our approach over the 7 Tomita Grammars[4], where each grammar defines the set of binary strings that should be accepted or rejected. Since, our focus is on policy learning problems, we treat the grammars as environments with two actions 'accept' and 'reject'. Each episode corresponds to a random string that is either part of the particular grammar or not. The agent receives a reward of 1 if the correct action accept/reject is chosen on the last symbol of a string.

**RNN Training.** The RNN for each grammar is comprised of a one-layer GRU with 10 hidden units, followed by a fully connected softmax layer with 2 nodes (accept/reject). Since we know the optimal policy, we again use imitation learning to train each RNN using the Adam optimizer and learning rate of 0.001. The training dataset is comprised of an equal number of accept/reject strings with lengths uniformly sampled in the range [1,50]. Table 2 presents the test results for the trained RNNs giving the accuracy over a test set of 100 strings drawn from the same distribution as used for training. Other than grammar #6[5], the RNNs were 100% accurate.

**MMN Training.** Since the raw observations for this problem are from a finite alphabet, we don't need to employ a bottleneck encoder to discretize the observation features. Thus the only bottleneck learned here is $b_h$ for the hidden memory state. We use the same architecture for $b_h$ as used for the MCE experiments and conduct experiments with $B_h \in \{8, 16\}$. These bottlenecks were then inserted in the RNNs to give MMNs. The performance of the MMNs before and after fine-tuning are shared in Table 2. In almost all cases, the MMN is able to maintain the performance of the RNN without fine-tuning. Fine tuning provides only minor improvements in other cases, which already are achieving high accuracy.

**Moore Machine Extraction.** Our results for MM extraction and minimization are in Table 2. In each case, we see a considerable reduction in the MM's state-space after minimization while accurately maintaining the MMN's performance. Again, this shows that the MMN learning does not directly result in minimal machines, yet are equivalent to the minimal machines and hence are exact

---

[4]The Tomita grammars are the following 7 languages over the alphabet $\{0, 1\}$: **[1]** $1^*$, **[2]** $(10)^*$, **[3]** any string without an odd number of consecutive 0's after an odd number of consecutive 1's, **[4]** any string not containing "000" as a substring, **[5]** all w for which #0(w) and #1(w) are even (where #a(w) is the number of a's in w), **[6]** any string such that the difference between the numbers of 1's and 0's is 3n , and **[7]** $0^*1^*0^*1^*$.

[5]Similar results are reported in Weiss et al. (2017)

Table 2: Moore Machine extraction for Tomita grammar

| #Grammar | RNN Acc. (%) | $B_h$ | Fine-Tuning Acc.(%) | | Before Minimization | | After Minimization | |
|---|---|---|---|---|---|---|---|---|
| | | | Before | After | $\left|\hat{H}\right|$ | Acc. (%) | $\left|\hat{H}\right|$ | Acc. (%) |
| 1 | 100 | 8 | 100 | - | 13 | 100 | **2** | 100 |
| | | 16 | 100 | - | 28 | 100 | **2** | 100 |
| 2 | 100 | 8 | 100 | - | 13 | 100 | **3** | 100 |
| | | 16 | 100 | - | 14 | 100 | **3** | 100 |
| 3 | 100 | 8 | 100 | - | 34 | 100 | **5** | 100 |
| | | 16 | 100 | - | 39 | 100 | **5** | 100 |
| 4 | 100 | 8 | 100 | - | 17 | 100 | **4** | 100 |
| | | 16 | 100 | - | 18 | 100 | **4** | 100 |
| 5 | 100 | 8 | 95 | 96 | 192 | 96 | 115 | 96 |
| | | 16 | 100 | - | 316 | 100 | **4** | 100 |
| 6 | 99 | 8 | 98 | 98 | 100 | 98 | 12 | 98 |
| | | 16 | 99 | 99 | 518 | 99 | 11 | 99 |
| 7 | 100 | 8 | 100 | - | 25 | 100 | **5** | 100 |
| | | 16 | 100 | - | 107 | 100 | **5** | 100 |

solutions. In all cases, except for grammar 6 the minimized machines are identical to the minimal machines that are known for these grammars (Tomita, 1982).

## 6 ATARI EXPERIMENTS

In this section, we consider applying our technique to RNNs learned for six Atari[6] games using the OpenAI gym (Brockman et al., 2016). Unlike the above experiments, where we knew the ground truth MMs, for Atari we did not have any preconception of what the MMs might look and how large they might be. The fact that the input observations for Atari (i.e. images) are much more complex than the previous experiments inputs makes it completely unclear if we can expect similar types of results. There have been other recent efforts towards understanding Atari agents (Zahavy et al., 2016; Greydanus et al., 2017). However, we are not aware of any other work which aims to extract finite state representations for Atari policies.

**RNN Training.** All the Atari agents have the same recurrent architecture. The input observation is an image frame, preprocessed by gray-scaling, 2x down-sampling, cropping to an $80 \times 80$ square and normalizing the values to [0, 1]. The network has 4 convolutional layers (kernel size 3, strides 2, padding 1, and 32,32,16,4 filters respectively). We used Relu as the intermediate activation and Relu6 over the last convolutional layer. This is followed by a GRU layer with 32 hidden units and a fully connected layer with $n+1$ units, where $n$ is the dimension of the Atari action space. We applied a softmax to first $n$ neurons to obtain the policy and used the last neuron to predict the value function. We used the A3C RL algorithm (Mnih et al., 2016) (learning rate $10^{-4}$, discount factor 0.99) and computed loss on the policy using Generalized Advantage Estimation ($\lambda = 1.0$) (Schulman et al., 2015). We report the trained RNN performance on our six games in the second column of Table 3.

**MMN Training.** We used the same general architecture for the QBN $b_f$ as used for the MCE experiments, but adjusted the encoder input and decoder output sizes to match the dimension of the continuous observation features $f_t$. For $b_h$, the encoder has 3 feed-forward layers with $(8 \times B_h), (4 \times B_h)$ and $B_h$ nodes. The decoder is symmetric to the encoder. For the Atari domains, the training data for $b_f$ and $b_h$ was generated using noisy rollouts. In particular, each training episode was generated by executing the learned RNN for a random number of steps and then executing an $\epsilon$-greedy (with $\epsilon = 0.3$) version of the RNN policy. This is intended to increase the diversity of the training data and we found that it helps to more robustly learn the QBNs. We trained bottlenecks for

---

[6]We use deterministic Atari games with frame-skip 4. Action Space of Pong and SpaceInvaders has been modified to [Noop, RightFire, LeftFire], [Noop, Fire, Right, Left], for the ease of training and interpretability.

$B_h \in \{64, 128\}$ and $B_f \in \{100, 400\}$ noting that these values are significantly larger than for our earlier experiments due to the complexity of Atari. Note that while there are an enormous number of potential discrete states for these values of $B_h$ the actual number of states observed and hence the number of MMN states can be substantially smaller. Each bottleneck was trained to the point of saturation of training loss and then inserted into the RNN to give an MMN for each Atari game.

**MMN Performance.** Table 3 gives the performance of the trained MMNs before and after fine-tuning for different combinations of $B_h$ and $B_f$. We see that for 4 games, Pong, Freeway, Bowling, and Boxing, the MMNs after fine tuning either achieve identical scores to the RNN or very close (in the case of boxing). This demonstrates the ability to learn a discrete representation of the input and memory for these complex games with no impact on performance. We see that for Boxing and Pong fine tuning was required to match the RNN performance. In the case of Freeway and Bowling, fine-tuning was not required.

In the remaining two games, Breakout and Space Invaders, we see that the MMNs learned after fine tuning achieve lower scores than the original RNNs, though the scores are still relatively good. On further investigation, we found that this drop in performance was due to poor reconstruction on some rare parts of the game. For example, in Breakout, after the first board is cleared, the policy needs to press the fire-button to continue, but the learned MMN does not do this and instead times out, which results in less score. This motivates the investigation into more intelligent approaches for training QBNs to capture critical information in such rare, but critically important, states.

**MM Minimization.** We see from Table 3 that before minimization, the MMs often have relatively large numbers of discrete states and observations. This is unsurprising given that we are using relatively large values of $B_h$ and $B_f$. However, we see that after minimizing the MMNs the number of states and observations reduces by orders of magnitude, sometimes to just a single state and/or single observation. The number of states and observations in many cases are small enough to write out and analyze by hand, making them amenable to careful analysis. However, this analysis is likely to be non-trivial for moderately complex policies due to the need to understand the "meaning" of the observations and in turn of the states.

**Understanding Memory Use.** We were surprised to observe in Atari the same three types of memory use considered for the MCE domains above. First, we see that the MM for Pong has just three states (one per action) and 10 discrete observation symbols (see Figure 2a). Most importantly we see that each observation transitions to the same state (and hence action) regardless of the current state. So we can view this MM as defining a set of rule that maps individual observations to actions with no memory necessary. In this sense, the Pong policy is analogous to the Amnesia MCE [5.1].

In contrast, we see that in both Bowling and Freeway there is only one observation symbol in the minimal MM. This means that the MM actually ignores the input image when selecting actions. Rather the policies are open-loop controllers whose action just depends on the time-step rather than the observations. Thus, these policies are analogous to the Blind MCE [5.1]. Freeway has a particularly trivial policy that always takes the Up action at each time step. While this policy behavior could have been determined by looking at the action sequence of the rollouts, it is encouraging that our MM extraction approach also discovered this. As shown in Figure 2b, Bowling has a more interesting open-loop policy structure where it has an initial sequence of actions and then a loop is entered where the action sequence is repeated. It is not immediately obvious that this policy has such an open-loop structure by just watching the policy. Thus, we can see that the MM extraction approach we use here can provide significant additional insight.

Breakout, Space Invaders and Boxing use both memory and observations based on our analysis of the MM transition structures. We have not yet attempted a full semantic analysis of the discrete observations and states for any of the Atari policies. This will require additional visualization and interaction tools and is an important direction of future work that is enabled by our approach.

## 7 Summary and Future Work

Motivated by the goal of better understanding memory use in RNN polices, we introduced an approach for extracting finite state Moore Machines from those policies. The key idea, bottleneck insertion, is to train *Quantized Bottleneck Networks* to produce binary encodings of continuous RNN memory and input features, and then insert those bottlenecks into the RNN. This yields a

Table 3: Moore Machine extraction for trained Atari RNN policies. DQN (Mnih et al., 2015), A3C (Mnih et al., 2016) scores have been reported for performance comparison with trained policies.

| Game (# of actions) | DQN (Score) | A3C LSTM (Score) | RNN (Score) | $(B_h, B_f)$ | Fine-Tuning Score | | Before Minimization | | | After Minimization | | |
|---|---|---|---|---|---|---|---|---|---|---|---|---|
| | | | | | Before | After | $|\hat{H}|$ | $|\hat{O}|$ | Score | $|\hat{H}|$ | $|\hat{O}|$ | Score |
| Pong (3) | 18.9 | 10.7 | **21** | 64,100 | 20 | 21 | 380 | 374 | 21 | 4 | 12 | **21** |
| | | | | 64,400 | 20 | 21 | 373 | 372 | 21 | 3 | 10 | **21** |
| | | | | 128,100 | 20 | 21 | 383 | 373 | 21 | 3 | 12 | **21** |
| | | | | 128,400 | 20 | 21 | 379 | 371 | 21 | 3 | 11 | **21** |
| Freeway (3) | 30.3 | 0.1 | **21** | 64,100 | 21 | - | 1 | 1 | 21 | 1 | 1 | **21** |
| | | | | 64,400 | 21 | - | 1 | 1 | 21 | 1 | 1 | **21** |
| | | | | 128,100 | 21 | - | 1 | 1 | 21 | 1 | 1 | **21** |
| | | | | 128,400 | 21 | - | 1 | 1 | 21 | 1 | 1 | **21** |
| Breakout (4) | 401.2 | 766.8 | 773 | 64,100 | 32 | 423 | 1898 | 1874 | 423 | 8 | 30 | 423 |
| | | | | 64,400 | 25 | 415 | 1888 | 1871 | 415 | 8 | 30 | 415 |
| | | | | 128,100 | 41 | 377 | 1583 | 1514 | 377 | 11 | 27 | 377 |
| | | | | 128,400 | 85 | 379 | 1729 | 1769 | 379 | 8 | 30 | 379 |
| Space Invaders (4) | 1976 | 23846 | 1820 | 64,100 | 520 | 1335 | 1495 | 1502 | 1335 | 8 | 29 | 1335 |
| | | | | 64,400 | 365 | 1235 | 1625 | 1620 | 1235 | 12 | 29 | 1235 |
| | | | | 128,100 | 390 | 1040 | 1563 | 1457 | 1040 | 12 | 35 | 1040 |
| | | | | 128,400 | 520 | 1430 | 1931 | 1921 | 1430 | 6 | 27 | 1430 |
| Bowling (6) | 42.4 | 41.8 | **60** | 64,100 | 60 | - | 49 | 1 | 60 | 33 | 1 | **60** |
| | | | | 64,400 | 60 | - | 49 | 1 | 60 | 33 | 1 | **60** |
| | | | | 128,100 | 60 | - | 26 | 1 | 60 | 24 | 1 | **60** |
| | | | | 128,400 | 60 | - | 26 | 1 | 60 | 24 | 1 | **60** |
| Boxing (18) | 71.8 | 37.3 | **100** | 64,100 | 94 | 100 | 1173 | 1167 | 100 | 13 | 79 | **100** |
| | | | | 64,400 | 98 | 100 | 2621 | 2605 | 100 | 14 | 119 | **100** |
| | | | | 128,100 | 94 | 97 | 2499 | 2482 | 97 | 14 | 106 | 97 |
| | | | | 128,400 | 97 | 100 | 1173 | 1169 | 100 | 14 | 88 | **100** |

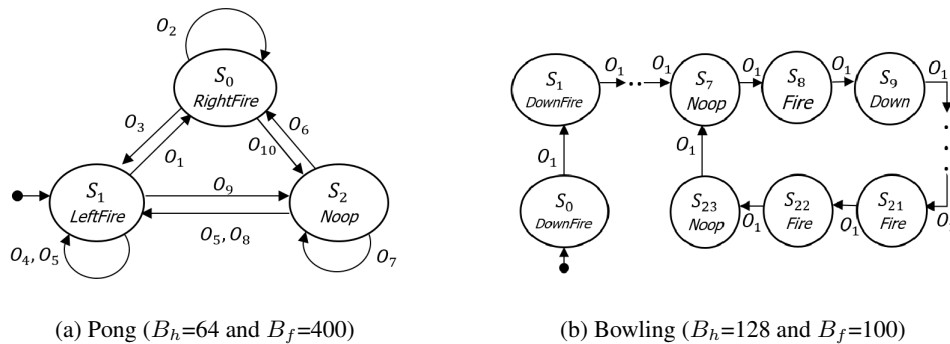

(a) Pong ($B_h$=64 and $B_f$=400)  (b) Bowling ($B_h$=128 and $B_f$=100)

Figure 2: Moore Machine representation for Atari policies

near equivalent Moore Machine Network (MMN) which has quantized memory and observation features. From the MMN we then extract a discrete Moore machine that can then be transformed into an equivalent minimal machine for analysis and usage. Our results on two environments where the ground truth machines are known show that our approach is able to accurately extract the ground truth. We also show experiments in six Atari games, where we have no prior insight into the ground truth machines. We show that, in most cases, the learned MMNs maintain similar performance to the original RNN policies. Further, the extracted machines provide insight into the memory usage of the policies. First, we see that the number of required memory states and observations is surprisingly small. Second, we can identify cases where the policy did not use memory in a significant way (e.g. Pong) and policies that relied only on memory and ignored the observations (e.g. Bowling and Freeway). To our knowledge, this is the first work where this type of insight was reported for policies in such complex domains. A key direction for future work is to develop tools and visualizations for attaching meaning to the discrete observations and in turn states, which will allow for an additional level of insight into the policies. It is also worth considering the use of tools for analyzing finite-state machine structure to gain further insight and analyze formal properties of the policies.

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

APPENDIX

7.1    FORMAL DEFINITION OF MODE COUNTER ENVIRONMENT

An MCE is parameterized by the mode number $M$, a mode transition function $P$, a mode life span mapping $\Delta(m)$ that assigns a positive integer to each mode, and a count set $C$ containing zero or more natural numbers. At time $t$ the MCE hidden state is a tuple $(m_t, c_t)$, where $m_t \in \{1, 2, \ldots, M\}$ is the current mode and $c_t$ is the count of time-steps that the system has been consecutively in mode $m_t$. The mode only changes when the lifespan is reached, i.e. $c_t = \Delta(m_t) - 1$, upon which the next mode $m_{t+1}$ is generated according to the transition distribution $P(m_{t+1} \mid m_t)$. The transition distribution also specifies the distribution over initial modes. The agent does not directly observe the state, but rather, the agent only receives a continuous-valued observations $o_t \in [0, 1]$ at each step, based on the current state $(m_t, c_t)$. If $c_t \in C$ then $o_t$ is drawn uniformly at random from $[m_t/M, (m_t + 1)/M)]$ and otherwise $o_t$ is drawn uniformly at random from $[0, 1]$. Thus, observations determine the mode when the mode count is in $C$ and otherwise the observations are uninformative. Note that the agent does not observe the counter. This means that to keep track of the mode for optimal performance the agent must remember the current mode and use memory to keep track of how long the mode has been active, in order to determine when it needs to "pay attention" to the current observation.

We conduct experiments with the following three MCE instances [7]: **1) Amnesia.** This MCE uses $\Delta(m) = 1$ for all modes, $C = \{0\}$, and uniformly samples random initial mode and transition distributions. Thus, an optimal policy will not use memory to track information from the past, since the current observation alone determines the current mode. This tests our ability to use MMN extraction to determine that a policy is purely reactive, i.e. not using memory. **2) Blind.** Here we use deterministic initial mode and transition distributions, mode life spans that can be larger than 1, and $C = \{\}$. Thus, the observations provide no information about the mode and optimal performance can only be achieved by using memory to keep track of the deterministic mode sequence. This allows us to test whether the extraction of an MMN could infer that the recurrent policy is ignoring observations and only using memory. **3) Tracker.** This MCE is identical to Amnesia, except that the $\Delta(m)$ values can be larger than 1. This requires an optimal policy to pay attention to observations when $c_t = 0$ and use memory to keep track of the current mode and mode count. This is the most general instance of the environment and can result in difficult problems when the number of modes and their life-spans grow. In all above instances, we used $M = 4$.

---

[7]Implementations available @ https://github.com/koulanurag/gym_x

## 7.2 GROUND TRUTH MCE MOORE MACHINES

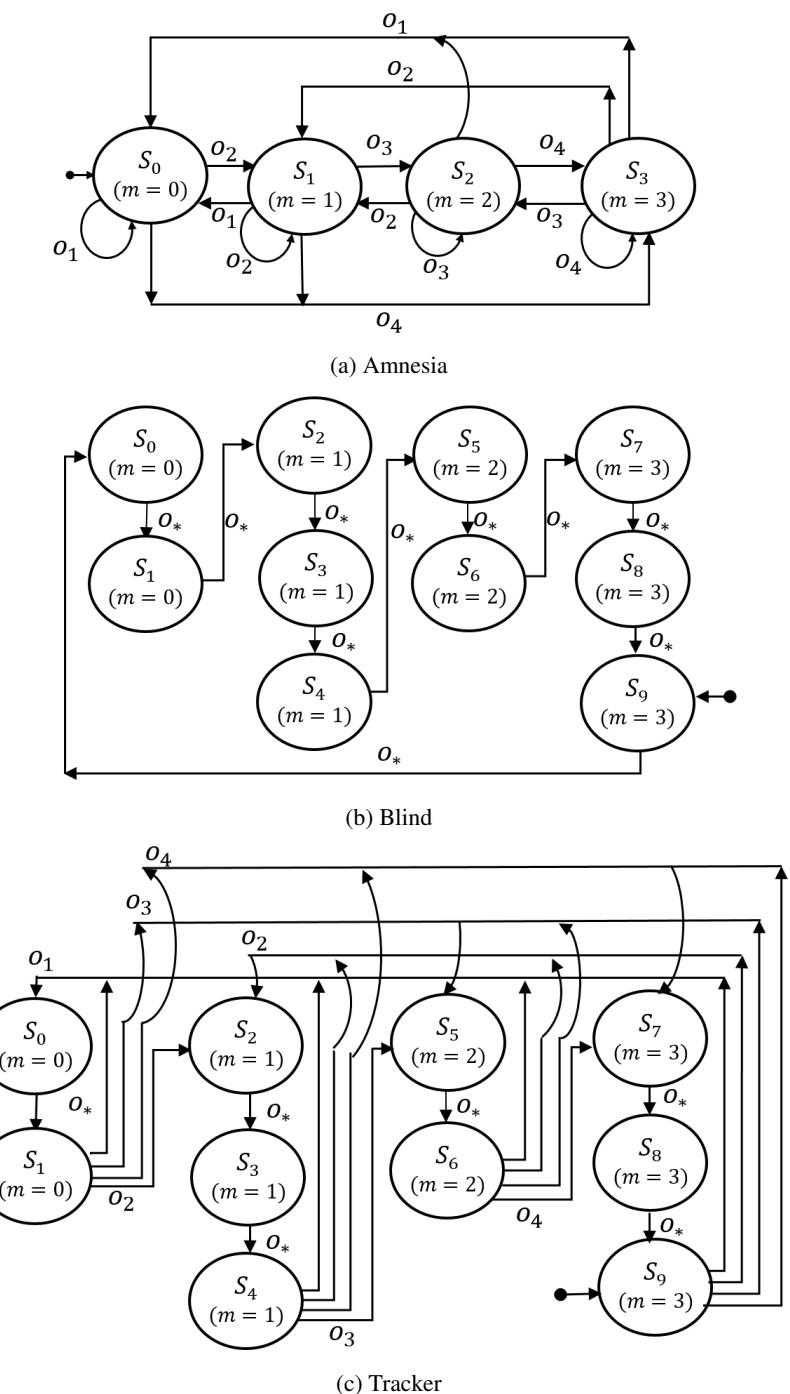

Figure 3: Moore machine representation of Mode Counter Environments (MCE). We use '$m$' to indicate the activate mode/action required in that state. Given $M = 4$, we have 4 observations classes, $o_1 = (0, 0.25]$, $o_2 = (0.25, 0.5]$, $o_3 = (0.5, 0.75]$ and $o_4 = (0.75, 1]$. Also, $o_*$ implies that the transaction is valid for all observations. The minimal moore machines extracted by our approach from trained RNN policies exactly matches these ground truth machines.

# 8 TOMITA MOORE MACHINES

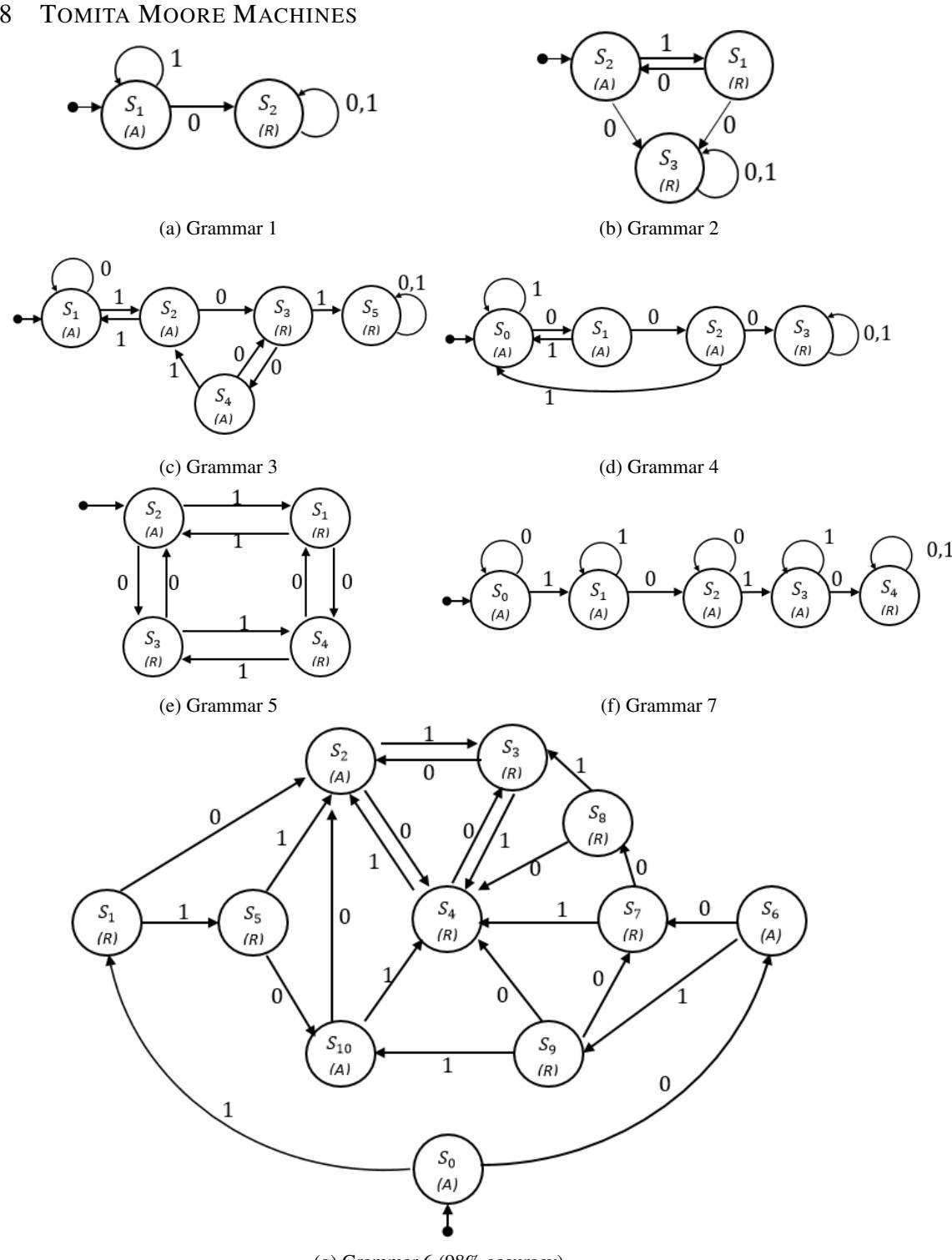

(a) Grammar 1

(b) Grammar 2

(c) Grammar 3

(d) Grammar 4

(e) Grammar 5

(f) Grammar 7

(g) Grammar 6 (98% accuracy)

Figure 4: Extracted Moore machine representation for Tomita Grammar policies where $B_h = 16$. We use 'A' and 'R' to denote accept and reject states, respectively. Other than Grammar 6, all machines are 100% accurate.

