# OpenReview forum: "Learning Finite State Representations of Recurrent Policy Networks"
_ICLR.cc/2019/Conference_

### Official Review · AnonReviewer1 · 2018-10-26
**An interesting work and need more comparisons with the most relative works**

**Rating:** 7
**Confidence:** 5

**Review:**

RNNs are difficult to explain, understand and analyze due to the continuous-valued memory vectors and observations features they use. Thus, this paper attempts to extract finite representation from RNNs so as to better interpret or understand RNNs. They introduce a new technique called Quantized Bottleneck Insertion to extract Moore Machines (MM). The extracted MM can be analyzed to improve the understanding of memory use and general behavior on the policies. The experiments on synthetic datasets and six Atari games validate the effectiveness of the proposal.

Here are my detailed comments:
Interpreting or understanding RNNs is a very interesting and important topic since RNNs and their variants like LSTM, GRU are widely used in different domains such as reinforcement learning, sentiment analysis, stock market prediction, natural language processing, etc. The more understandable on RNNs, the more trustful on them. In this paper, the authors try to extract more interpretable representation of RNNs, namely Moore Machines (MM). MM is actually a classical finite state automaton. The authors mention that (Zeng et al., 1993) is the most similar work to theirs. In fact a series of works have been proposed to extract finite state automaton, which is similar to (Zeng et al., 1993) such as [1], [2], [3], etc. I think the authors could make the related works more complete by incorporating these literatures I mentioned.

Besides, I think this work is a good application of the idea of extraction of RNNs on reinforcement learning since no works have introduced this idea into this domain as far as I know. The authors use the autoencoder named as QBN to quantize the space of hidden states. This is a good operation of clustering or quantizing the space of hidden states since it can be tuned to make the final performance better. The authors also incorporate the minimization of MM to show the probability of shrinking memory which can also make the extracted MM more interpretable. As a result, the policy represented by MM is intuitive and vivid.

Nevertheless, there is an obvious weak point in this paper. Specifically, the authors claim that the main contribution of this paper is to introduce an approach for transforming RNNs to finite state representations. But I do not see any comparisons between the proposed methods and other relative methods such as the method proposed by (Zeng et al., 1993) to show the effectiveness or improvement of the proposed method. I suggest the authors could incorporate comparisons to make the results more convincing.

[1] C. W. Omlin and C. L. Giles, "Extraction of rules from discrete-time recurrent neural networks," Neural Networks, vol. 9, no. 1, pp. 41–52, 1996.
[2] C. W. Omlin and C. L. Giles, "Constructing deterministic finite-state automata in recurrent neural networks," Journal of the ACM, vol. 43, no. 6, pp. 937-972, 1996.
[3] A. Cleeremans, D. Servan-Schreiber, and J. L. McClelland. "Finite state automata and simple recurrent networks." Neural computation, vol. 1, no. 3, pp. 372-381, 1989.

---

> ### Author Response · Authors · 2018-11-10
> **Response to Review**
>
> Thanks for the comments. Below we pull quotes from the review followed by responses.
>
> "the authors could make the related works more complete by incorporating these literatures I mentioned."
>
> RE: Indeed the literature on FSM extraction is quite vast and we tried to include representative papers from the different classes of approaches. We will be happy to include the papers you pointed to, noting that these are just a few from this class of approaches that have appeared over the years.
>
> "obvious weak point … do not see any comparisons between the proposed methods and other relative methods such as the method proposed by (Zeng et al., 1993)"
>
> RE: As we mentioned in the related work, there is no prior work that we are aware of that attempts to learn to transform RNNs into Moore Machines. We would be happy to get pointers to related work that we can compare with.
>
> We included a discussion of work on learning FSMs in the related work, because those techniques are related to our problem. But NONE of the approaches that we are aware of can be applied to our problems without significant innovation.  This is due to two reasons: 1) Our inputs are complex objects (images or real numbers) compared to FSM learning where the inputs are from a discrete alphabet, and 2) FSMs are different from Moore Machines, since Moore Machines must output an action/symbol at each time step, rather than just accepting/rejecting entire strings as is the case for FSMs. So FSM approaches are not directly applicable.
>
> For the Grammar Learning benchmarks, prior FSM methods can apply (since actions are just accept/reject). However, here we achieve nearly perfect performance, so a comparison would not shed additional light.

---

> > ### Comment · AnonReviewer1 · 2018-11-11
> > **There is no essential difference between Moore Machines and Finite State Machines**
> >
> > The authors mentioned that FSMs are different from Moore Machines (MMs), since Moore Machines must output an action/symbol at each time step, rather than just accepting/rejecting entire strings as is the case for FSMs. In my opinion, there is no essential difference between MMs and FSMs. The main reason is that in FSMs, the accepting/rejecting state reflects a binary classification scenario while the actions/symbols output in MMs reflects a multi-class classification scenario. The classical binary FSMs can be easily adapted to the multi-class classification version.
> >
> > Besides, the authors stress that the input of MMs are complex objects including images or real numbers while FSM can only learn from a discrete alphabet. This is not an issue since in this paper the authors firstly use CNN to encode the complex input into a simple form that MMs can accept, which is similar with the input that FSMs can accept.
> >
> > The key idea of this paper is to discretize the hidden states and thus the similar hidden states can be grouped together to form a state representing an action. The main contribution is that the authors bring external CNN to encode the complex input into a form that MMs can accept and use a new technique called QBN to do discretization or clustering and apply this idea to the reinforcement learning tasks. There is no essential difference between this paper and the large number of literatures on extracting FSMs from RNNs. Thus, the comparisons between them are feasible with some adaptations on the classical ones.

---

> > > ### Author Response · Authors · 2018-11-12
> > > **Quantizing RNNs for complex games like Atari is not as simple as the reviewer suggests**
> > >
> > > The reviewer states:
> > >
> > > "The classical binary FSMs can be easily adapted to the multi-class classification version. ….. There is no essential difference between this paper and the large number of literatures on extracting FSMs from RNNs."
> > >
> > > The reviewer's criticism is that there are simple extensions to prior FSM learning techniques that could be used to achieve our results, even the Atari results.
> > >
> > > This is a vague hypothesis that requires a careful argument.  We give a technical analysis of the hypothesis at the end of the response by detailing our unsuccessful experience with “simple extensions”. First we give a higher level response.
> > >
> > > Discounting our contributions by requiring comparison to unspecified and non-existent extensions is unfair.  Note that 2 of the 3 references suggested by the reviewer [2,3] do not involve extracting FSMs from RNNs (see technical response).
> > >
> > > Such extensions are not as straightforward as the reviewer implies.  This is a very different from not comparing to a clearly specified off-the-shelf approach or a small change to such an approach.
> > >
> > > Finally, we hope the reviewers recognize the contribution of demonstrating finite-state extraction for problems as complex as Atari. We were very surprised by the success on Atari, that the state spaces were so small, and that we could determine that some policies did not use memory and that some policies only used memory.
> > >
> > > ---------------------------------------
> > > Detailed Technical Response:
> > > ---------------------------------------
> > >
> > > We considered and tried a variety of approaches, starting with "simple" extensions. We would have been happy for any of these to work and would have written about the result. However, failures to get such approaches to work led to our proposed QBN-insertion approach.
> > >
> > > Let us examine the simple extensions that the reviewer might be considering. We'll divide the discussion into three parts.
> > >
> > > 1) *Reviewer Suggested References [2] and [3]*
> > >
> > > [1,2] are not about extracting FSMs from RNNs. [2] starts with an FSM and compiles it into an RNN as prior knowledge for bootstrapping. [3] trains RNNs on FSM languages and analyzes their ability to do this. They do not give a method for extracting FSMs.
> > >
> > > 2) *Reviewer Suggested Reference [1]: post-gradient descent clustering/discretization*
> > >
> > > [1] falls in the class of approaches in “related work” (para 2). It trains an RNN, clusters the internal states (discretization or k-means), then connect states based on empirical data. The resulting FSM is disconnected from the original RNN. We have only seen these approaches applied to relatively simple problems with a small number of discrete inputs. More importantly, when the approach does not give an accurate FSM, there is no easy way to fine tune because it is disconnected from the original network. As our tables show, fine-tuning was essential to achieve good performance on more difficult problems.
> > >
> > > A simple extension, for problems such as Atari, is to cluster the continuous representation of the input (e.g. output features of CNN). This was the first approach that we tried and were unable to get good results for all but the smallest problems, despite serious attempts. Results were better than random for larger problems, but there is no way to further improve via fine tuning. Thus, we are skeptical that there is a simple extension.
> > >
> > > QBN-insertion gives a method for getting the required clusterings in a way that can be directly embedded in the RNN for fine-tuning.
> > >
> > > 3)  *(Zeng et al., 1993): training binary RNNs from scratch*
> > >
> > > This work defines an RNN with discretized memory and trains from scratch. It has not been applied to large problems or to RL.
> > >
> > > A simple extension would also discretize the input (e.g. by discretizing the output features of a CNN). This was our second attempt. As mentioned in the paper, we did not get it to work for large problems. Training from scratch for Atari was unsuccessful after days. Apparently the discretized nodes make it difficult for RL training signals to effectively propagate. We experimented with some more recent techniques for learning with discrete units, without success. Thus, we are skeptical that there is a simple extension. The failures are mentioned in the paper, but we will make the results more prominent.
> > >
> > > These failures led us to the QBN-insertion approach followed by fine-tuning.

---

> > > > ### Comment · AnonReviewer1 · 2018-11-12
> > > > **QBN and fine-tuning are effective ways to make the method scalable**
> > > >
> > > > Thanks for pointing out the effectiveness of QBN insertion and fine-tuning. I realize there is a bigger contribution than I thought before. I will change my score from 6 to 7.

---

### Official Review · AnonReviewer2 · 2018-10-30

**Rating:** 7
**Confidence:** 3

**Review:**

This paper proposes a method to learn a quantization of both observations and hidden states in an RNN. Its findings suggest that many problems can be reduced to relatively simple Moore Machines, even for complex environments such as Atari games.

The method works by pretraing an RNN to learn a policy (e.g. through the A3C algorithm), and then training pairs of encoder/decoder networks with a quantizing forward pass and a straight-through backpropagation. The learned quantizations can then be used to build a Moore Machine, which itself can be reduced with FSM reduction algorithms, yielding a discrete, symbolic approximation of the inner workings of RNNs, that could in principle be interpreted more easily than latent embedding spaces.

One downside of this paper is that it promises an exciting method to analyse the inner workings of RNNs, but then postpones this analysis to later work. Understandably, the synthetic experiments take some space and shows that the proposed method works as expected when the problem is amenable to discretization; maybe some parts of this could be in the appendix?

Another downside is that there is little indication of the computational implications of the method. The method was evaluated on a fairly small set of hyperparameters, and there are no indication of how long the optimization and finetuning takes. Presumably, minimizing a Moore Machine has been studied for decades, but how long does minimizing the 1000s of states in Atari games take? A second or an hour?

The paper is fairly well written and easy to understand. The method seems well grounded, although I'm not familiar enough with the quantization literature to detect if something important is missing. I think this is a great tool that hopefully will be used to try to understand the memory mechanisms of RNNs.

I think the proposed method (and the fact that it works in simple cases) warrants acceptance, but I think more experimental work would make this a great contribution. Since there is no reason for quantization to improve performance if it is done after training, then more emphasis should be put on the interpretability of the discretization; yet it is lacking in the current work. Some Atari games are known to require various amounts of memory, this could be analysed. Some other Atari games are known to be hard to solve, what happens to the RNN when the agent fails to achieve an optimal policy might also show up in the subsequent discretization and be interesting to analyse.

Comments:
- In atari, you can have access to the RAM and from it, using exactly the same mechanisms and maybe a bit of tabular MDPs, you should be able to recover the optimal MM.
- It is good that the authors report their failure to train MMNs from scratch; IMO this says something about the straight through estimators' limits. Measuring how sensible these things are to change in their target distribution and comparing to previous uses of ST in quantization works could be interesting.
- in Section 8 (appendix) "Grammer" should be "Grammar"
- All the (PO)MDPs that you analyse arguably have finite state spaces, and you set the ALE to be deterministic. What happens in continuous stochastic environments?
- Do you think a similar technique could be used to recover a (possibly stochastic) MDP instead of a Moore Machine? It would be interesting to see MDP reduction methods applied to a learned MDP.

---

> ### Author Response · Authors · 2018-11-10
> **Response to Review**
>
> Thanks for the comments. Below we pull quotes from the review followed by responses.
>
> "One downside of this paper is that it promises an exciting method to analyse the inner workings of RNNs, but then postpones this analysis to later work."
>
> RE: First, we do want to point out that while we were not able to fully analyze the Moore Machines resulting from Atari games, we were able to make some interesting observations about the memory use, which we've never seen done before (last part of Section 6). In particular, some policies did not even use observations and others did not really use memory. We don't know of any prior technique that could be employed to uncover these surprising aspects of the policies.
>
> We agree that it would have been ideal to provide a full analysis of the Atari policies using additional visualization tools. However, even without that, the work has taken the big step of showing how to create quantized representations that would allow such an analysis. It was very surprising to us that this was able to work for Atari games at all.
>
> "little indication of the computational implications of the method … how long does minimizing 1000s of states in Atari games take"
>
> RE: We will provide some information on optimization times in the revised paper (Will be uploaded soon.) We can say now that training the quantized autoencoders + fine tuning is faster than training a Moore Machine Network from scratch (which failed for Atari games) and much faster than training the original RNN policy.
>
> Minimization is quite fast. For the largest numbers of states in Atari the minimization took a couple of minutes, noting that this was highly unoptimized minimization code. The synthetic problems took seconds to minimize.
>
> "In atari, you can have access to the RAM and from it, using exactly the same mechanisms and maybe a bit of tabular MDPs, you should be able to recover the optimal MM."
>
> RE: Not sure what you have in mind here. It seems unlikely that we would be able to get an optimal MM for an Atari game even with the use of RAM. The problem is just too large to solve optimally.
>
> "All the (PO)MDPs that you analyse arguably have finite state spaces, and you set the ALE to be deterministic. What happens in continuous stochastic environments?"
>
> RE: The POMDPs we use actually have continuous observations (but yes finite states) with some stochastic behavior in the observation generation. We haven't used policies trained on stochastic variants of ALE, but agree this is an interesting direction to consider.
>
> "Do you think a similar technique could be used to recover a (possibly stochastic) MDP instead of a Moore Machine? It would be interesting to see MDP reduction methods applied to a learned MDP."
>
> RE: This is an interesting idea and it does seem possible. The "bottleneck insertion" approach is quite general and can be plugged into a network at any point where quantization seems useful to introduce.

---

> > ### Comment · AnonReviewer2 · 2018-11-12
> > **More comments**
> >
> > It's interesting that training the quantization is faster than the original model, I didn't expect it to be so.
> >
> > It is true that solving Atari games from RAM might require resources you do not have. Alternatively, you could simply run your already trained agents, collect RAM from those trajectories, and then perform a similar analysis from the RAM states instead of the quantized hidden states. This could reveal if the RNN remembers more than there is to remember or alternatively if it ignores certain parts of the true state space which might be mostly unnecessary.
> >
> > I think one simple experiment that could be done w.r.t. a remark by Reviewer3 is to design a simple POMDP where the optimal strategy requires counting a real-valued number. In such a case I assume your method would fail, but if it doesn't, that might be even more interesting.
> >
> > In that respect, one thing that would make me change my score from 6 to 7 is if the authors can convince me that their method could be used to inform hyperparameter search for RNNs. Is this something you have observed empirically while building experiments for this paper?

---

> > > ### Author Response · Authors · 2018-11-14
> > > **Response to "More Comments"**
> > >
> > > Regarding:
> > >
> > > "It's interesting that training the quantization is faster than the original model, I didn't expect it to be so."
> > >
> > > There are two likely reasons for this.
> > >
> > > First, supervised learning is generally easier than reinforcement learning. For example, training an RNN for Atari games can take several hours with lots of parallelism, or many hours without parallelism. Rather, given a trained RNN, the training of quantized autoencoders (QBNs) is effectively a supervised problem (minimize reconstruction error) and then fine-tuning is also a supervised problem (mimic the original RNN).
> > >
> > > Second, learning an RNN from scratch for challenging problems, whether it is supervised learning or reinforcement learning, is significantly more challenging than just learning the quantized autoencoders (which avoids BP through time) and then fine-tuning (starts from a good place in parameter space).
> > >
> > > Regarding,
> > >
> > > "This could reveal if the RNN remembers more than there is to remember or alternatively if it ignores certain parts of the true state space which might be mostly unnecessary."
> > >
> > > In Atari, the learned MMs are certainly remembering much less than the full state of the game. The extreme case is Pong, where, as described in the paper, the MM does not use any real memory (see "Understanding Memory Use" on pg. 9). Table 3 also shows that most of the time the number of MM states is less than 10 and at most 33 in our experiments. So this would translate to about 5 bits of memory compared to the 128 bit Atari RAM.
> > >
> > > Regarding,
> > >
> > > "design a simple POMDP where the optimal strategy requires counting a real-valued number"
> > >
> > > For problems where significant counting, or more generally, value accumulation is required, then Moore Machines are probably not the best type of representation to extract. In such cases, models such as Petri Nets might be more appropriate.
> > >
> > > In general, there are a variety of qualitatively different ways that memory can be used in a recurrent system. We expect that there will be an interesting line of research focused on understanding the primary usage classes and developing corresponding extraction approaches (e.g. Petri Nets). It may be the case, that the QBN-insertion approach here can be a schema for such developments (<your favorite structure>-insertion).
> > >
> > > Regarding,
> > >
> > > "make me change my score from 6 to 7 is if the authors can convince me that their method could be used to inform hyperparameter search for RNNs"
> > >
> > > Naturally, we would love to say something useful here, but are not exactly sure what the reviewer has in mind.
> > >
> > > Our extraction approach is applied after an RNN has been learned. So in that sense, it would not directly inform hyperparameter search for RNNs if applied directly.
> > >
> > > We have observed in some cases that when a trained RNN R1 does not generalize as well as another trained RNN R2, that the MM extracted for R2 is more compact than that for R1. For example, see Grammar #5 in Table 2. When using 8 versus 16 memory bits we get 96 versus 100 percent accuracy. The corresponding minimized MMs have 115 versus 4 states.
> > >
> > > This provides a bit of evidence that given two RNNs that perform similarly on validation data, we might prefer to use the one that results in a more compact MM. This makes some intuitive sense, but is at best a hypothesis at this point.

---

### Official Review · AnonReviewer3 · 2018-11-01
**Interesting in terms of interpretability but unclear practical advantage wrt state of the art**

**Rating:** 6
**Confidence:** 3

**Review:**

Approximation of RNNs is a hot and important topic in term of interpretability and control of nets. The related work section is good but in my opinion miss to give a position with respect to the work dedicated to extract rules from a net which are also way to "interpret" a RNNs - as an example https://arxiv.org/abs/1702.02540 from ICLR'17.

pros:
- important practical topic
- The papers includes a variety of ideas/tricks which seems to bring performance as the 3 stage procedure and the gradient backpropagation over quantization.
- Makes "interpretable" observations of some no so easy to understand nets on Atari games
- Reach state of the art performance on artificial set of task

cons:
- The impact of each step is not always assessed by an experiment (especially ones introduced in section 4.1)
- The method is never benchmarked against an other one. Neither in terms of performance of the approximation nor in terms of interpretability (thought other techniques are cited in the paper). I understand that this is because this pursue the two goals at the same time but I'd be interested this tradeoff to be more investigated.
- Performance on Atari games is usually reported in term of % wrt human performance which helps understanding where we stand. It would be good also to discuss the performance of the RNN on the game wrt other nets. As an example in this paper on space invaders the performance of the RNN is slightly better human but very far from state of the art yielded by prioritized duelling which is almost 10x higher in terms of score. While on breakout they are very good (see https://arxiv.org/pdf/1806.06923.pdf to have a recent list of score on Atari).
- I'd been interested in having an artificial task where to proposed algorithm does not succeed (an ideally some discussion on what make the structure recoverable or not).

---

> ### Author Response · Authors · 2018-11-10
> **Response to Review**
>
> Thanks for your time and comments. Below we pull quotes from the review followed by responses.
>
> "miss to give a position with respect to the work dedicated to extract rules …. example  https://arxiv.org/abs/1702.02540"
>
> RE: We will add rule-extraction techniques as a related class of methods to related work. (Revision will be coming.) However, we have not seen a rule-extraction approach that can be easily adapted to our target problems where the recurrent policies consume complex inputs such as images (Atari experiments) or real-valued inputs (mode counter experiments). So while related they are not really competing approaches without significant innovation.
>
> "Impact of each step is not always assessed …. especially ones introduced in section 4.1"
>
> RE: The two choices not assessed were: 1) The level of quantization, where we use three levels {-1, 0, 1}, and 2) The impact of using the "flattened tanh" function.
>
> For (1) we have not done an in depth comparison of the impact of # of quantization levels. This is primarily due to the expense of producing the comparison. As mentioned in the paper, this is an interesting point to explore in the future, but was not really central to our main goal of demonstrating the potential for this new approach.
>
> For (2) during early experimentation we did some comparison to using pure tanh and observed a small advantage to using the flattened version. At that point, due to experimental cost, we needed to stick to one or the other and chose the flattened tanh. We may be able to include limited comparisons in the revision, but it will need to be in the appendix for space reasons.
>
> "never benchmarked against an other one. Neither in terms of performance of the approximation nor in terms of interpretability"
>
> RE: As we mentioned in the related work, there is no prior work that we are aware of that attempts to learn to transform RNNs into Moore Machines. We would be happy to get pointers to related work that we can compare with.
>
> We included a discussion of work on learning FSMs in the related work, because those techniques are related to our problem. But NONE of the approaches that we are aware of can be applied to our problems without significant innovation.  This is due to two reasons: 1) Our inputs are complex objects (images or real numbers) compared to FSM learning where the inputs are from a discrete alphabet, and 2) FSMs are different from Moore Machines, since Moore Machines must output an action/symbol at each time step, rather than just accepting/rejecting entire strings as is the case for FSMs. So FSM approaches are not directly applicable.
>
> For the Grammar Learning benchmarks, prior FSM methods can apply (since actions are just accept/reject). However, here we achieve nearly perfect performance, so a comparison would not shed additional light.
>
> "Performance on Atari games is usually reported in term of % wrt human performance … discuss the performance of the RNN on the game wrt other nets"
>
> RE: It is important to recall the primary goals of this paper. We are NOT trying to train the best Atari playing policies. Rather, our aim is to study how to create finite state representations for problems as complex as Atari games. In this sense, normalizing scores and/or comparing performance to other Atari playing policies is orthogonal to our goals. Rather it is primarily important to indicate that we are dealing with policies that are achieving reasonable performance.
>
> To address your concern and give a point of reference for our policy qualities, we will report the Nature DQN and current SOTA scores for those games in the next draft.  Although some researchers tend to report scores w.r.t. human baselines, there is a fair bit of disagreement about where those baselines should lie (Figure 1 in http://gershmanlab.webfactional.com/pubs/Tsividis17.pdf seems to indicate, for example, that the DeepMind baselines are too low). So we prefer just to report raw scores for now.
>
> "interested in an artificial task where the algorithm does not succeed"
>
> RE: The third paragraph on page 9 gives 2 examples from Atari where the approach results in a decrease in performance after discretization. We give our best explanation for why this happens there. We agree that it will be interesting future work to design classes of artificial problems, where different complexity parameters can be modified for testing the limits of our approach.

---

### Author Response · Authors · 2018-11-23
**Changes log**

Thanks to all reviewers for their comments and suggestions.

We have made slight changes in the paper:
	[Reviewer 1] - Cited Rule Extraction Work in Related Work Section.
	[Reviewer 1] - Added DQN, A3C scores in Table 3 for policy performance comparison with trained Atari policies.
	[Reviewer 2] - Corrected "Grammar" spelling in Appendix.

---

### Meta-Review · Area_Chair1 · 2018-12-14
**Accept**

**Confidence:** 4
**Recommendation:** Accept (Poster)

**Metareview:**

The paper addresses the problem of interpreting recurrent neural networks by quantizing their states an mapping them onto a Moore Machine. The paper presents some interesting results on reinforcement learning and other tasks. I believe the experiments could have been more informative if the proposed technique was compared against a simple quantization baseline (e.g. based on k-means) so that one can get a better understanding of the difficulty of these task.

This paper is clearly above the acceptance threshold at ICLR.